# Improving Audiology Student Training by Clinical Simulation of Tinnitus: A Glimpse of the Lived Experience of Tinnitus

**DOI:** 10.3390/brainsci13091338

**Published:** 2023-09-17

**Authors:** Pierre H. Bourez, Guillaume T. Vallet, Philippe Fournier

**Affiliations:** 1Department of Rehabilitation, Laval University, Quebec City, QC G1V 0A6, Canada; 2Centre for Interdisciplinary Research in Rehabilitation and Social Integration (CIRRIS), Centre Intégré Universitaire de Santé et de Services Sociaux de La Capitale-Nationale (CIUSSS-CN), Quebec City, QC G1M 2S8, Canada; 3Department of Psychology, Université du Québec à Trois-Rivières, Trois-Rivières, QC G9A 5H7, Canada; guillaume.vallet@uqtr.ca; 4Centre de Recherche de L’institut Universitaire de Gériatrie de Montréal, Montreal, QC H3W 1W4, Canada

**Keywords:** tinnitus, clinical simulation, audiology training, psychoacoustic measures

## Abstract

Purpose: Student audiology training in tinnitus evaluation and management is heterogeneous and has been found to be insufficient. We designed a new clinical simulation laboratory for training students on psychoacoustic measurements of tinnitus: one student plays the role of the tinnitus patient, wearing a device producing a sound like tinnitus on one ear, while another student plays the role of the audiologist, evaluating their condition. The objective of the study was to test this new clinical simulation laboratory of tinnitus from the perspective of the students. Method: This study reports the findings from twenty-one audiology students (20 female and 1 male, mean age = 29, SD = 7.7) who participated in this laboratory for a mandatory audiology class at the Laval University of Quebec. Three students had hearing loss (one mild, two moderate). All students played the role of both the clinician and the patient, alternately. They also had to fill out a questionnaire about their overall experience of the laboratory. Results: The qualitative analysis revealed three main themes: “Benefits of the laboratory on future practice”, “Barriers and facilitators of the psychoacoustic assessment”, and “Awareness of living with tinnitus”. The participants reported that this experience would have a positive impact on their ability to manage tinnitus patients in their future career. Conclusion: This fast, cheap, and effective clinical simulation method could be used by audiology and other healthcare educators to strengthen students’ skills and confidence in tinnitus evaluation and management. The protocol is made available to all interested parties.

## 1. Introduction

Tinnitus is defined as an auditory sensation without an external sound stimulation or meaning [1]. It is usually described as a constant hissing, ringing or buzzing sound heard in one ear, in two ears, or in the head [2]. The prevalence of tinnitus is estimated globally to be close to 15–20% of the general population [3,4,5]. The prevalence of tinnitus usually increases with age, with a higher prevalence for older age groups. It is also known that tinnitus may occur at earlier stages in life, such as in children and adolescents [6]. However, two recent Canadian studies revealed either a similar prevalence of tinnitus across age groups [7] or a slightly increased prevalence for younger adults [8]. The first study revealed that more than one in two Canadians reported having experienced at least occasional tinnitus in the last year (8% all the time, 10% often, 37% occasionally; BIP recherche, 2022). The second study reported an overall prevalence of 37% among the Canadian population for experiencing tinnitus in the last year, with 7% reporting that it was bothersome [8]. It has been reported that many individuals may hear more than one tinnitus sound [2,9,10] and that multiple types of tinnitus can interact and reside within a single patient [11]. Tinnitus can be lived as an unpleasant experience, possibly impacting the quality of life [1], with the most well-known complaints being sleep and concentration problems, listening difficulties, anxiety, isolation, and depression [12]. The impact of tinnitus on quality of life is usually assessed using various standardized questionnaires such as the Tinnitus Handicap Inventory (THI) [13] and the Tinnitus Functional Index (TFI) [14]. To date, cognitive behavioral therapy has been shown to be the most effective treatment for tinnitus [15].

Audiologists generally play a central role in the assessment and management of people with tinnitus, at least in North American countries such as Canada and the United States. Most tinnitus patients display some level of hearing loss, requiring a complete audiological evaluation to investigate potential underlying causes [16]. Sound therapies including amplification and various other forms of sound stimulation designed to interfere with the tinnitus perception and/or reaction are usually provided by audiologists [17], highlighting their central role. The role of the audiologist is significant in tinnitus evaluation and management, but should be integrated within a widely preferred and supported multidisciplinary approach, including the involvement of various healthcare professionals such as psychologists, otolaryngologists, psychiatrists and physiotherapists, to name a few [18]. Still, it is now well accepted that tinnitus evaluation and management are professional acts that are within the scope of practice of audiologists, as advocated by national professional organizations such as the American Speech and Hearing Association and the American Academy of Audiology (USA), Speech and Audiology Canada and the Canadian Academy of Audiology (Canada). However, the level of knowledge regarding tinnitus evaluation and care shown by audiologists may vary greatly from one professional to another [19]. One potential factor contributing to this heterogeneity may be the differences in training devoted to tinnitus across audiology programs. In the United States, audiology training programs are accredited by the Council on Academic Accreditation in Audiology (CAA) and Speech–Language Pathology of the American Speech–Language and Hearing Association (ASHA). This accreditation system is designed to ensure that audiology programs provide adequate training. Tinnitus was incorporated into the standard requirements only in August 2017. Before this date, tinnitus evaluation and management were not always covered by university programs [20]. These new standards now include the identification and prevention of tinnitus, the assessment of tinnitus (perform an assessment to characterize tinnitus and provide counseling in a culturally sensitive manner) and the intervention that minimizes the effect of tinnitus (perform the assessment for tinnitus intervention and assess the efficacy of tinnitus intervention). A recent study surveyed 32 audiology training programs in the USA and found that tinnitus training was highly variable from one program to another [20]. Audiology programs were asked about their confidence in their students’ ability to provide (1) an appropriate referral for a medical evaluation due to tinnitus symptoms, (2) an appropriate assessment of how tinnitus affects a patient functionally and emotionally, (3) counseling for bothersome tinnitus, and (4) specific tinnitus interventions. The lowest confidence scores were found for intervention and counseling. This is not surprising, as it can be difficult for a student to fully understand the subjective and highly heterogeneous nature of tinnitus, and its impact on the life of a patient. Although it is conceivable that audiology students may have experienced temporary tinnitus in their lives [21], in our experience, only a minority of audiology students have experienced chronic tinnitus lasting several minutes or even hours. Therefore, they have no lived experience of tinnitus on which to base their theoretical and practical knowledge.

Simulation-based education has been widely used in the healthcare education field to address such concerns. Clinical simulation can be defined as “a technique—not a technology—to replace or amplify real experiences with guided experiences that evoke or replicate substantial aspects of the real world in a fully interactive manner” [22]. Clinical simulation can be used for both evaluating clinical competencies and/or as an experiential learning opportunity for students [23]. Simulations have been adopted as a teaching method in several health-related disciplines, such as medicine [24], nursing [25] and physiotherapy [26]. Clinical simulations were shown to be associated with better learning outcomes, including knowledge, skills, and behavior, in a meta-analysis [27]. On a further note, simulation can also be used as a method for recreating what it is like to be a patient by simulating the features of the experience of illness. As stated in a scoping review [28], these approaches aim to encourage critical reflection through experience and may cultivate greater empathic care towards patients. Despite these significant advantages, clinical simulation is not widely used by audiology and speech–language pathology educators, as reported in a recent study [23], and to the best of our knowledge, it has never been used for teaching tinnitus evaluation and management.

The current article aims to explore the benefit of a new simulated tinnitus patient approach in teaching audiology students the psychoacoustic assessment of tinnitus by directly questioning them about their overall experience of the laboratory. We developed a training protocol in which each student successively plays the role of the audiologist and of the tinnitus patient. The student playing the role of the patient is asked to wear, in one ear, a device that produces a constant sound similar to tinnitus (high-frequency pure tone played at a few dB SL). The student playing the role of the audiologist is required to perform the four typical psychoacoustic measures of tinnitus, that is, tinnitus pitch (1) and loudness matching (2), the minimum masking level or MML (3), and the residual inhibition or RI (4) [29,30]. Tinnitus pitch and loudness matching consists of estimating the frequency and loudness of the tinnitus while using an external sound. The MML consists of measuring the minimum level of a sound or a noise required to mask tinnitus. Finally, the RI measurement consists of measuring whether the presentation of sound can temporarily suppress tinnitus. After it is completed, the two students switch their roles: the student playing the patient becomes the clinician and vice versa. We believe that this approach is valuable for developing psychoacoustic testing skills (simulated audiologist) as well as developing a better comprehension of the lived experience of tinnitus (simulated tinnitus patient). Confronting them with the perception of intrusive constant noise, and having them describe a very subjective experience and undergo the same type of test as their future patient may help them get a better sense of what it “feels like” to have tinnitus and be evaluated by an audiologist. Of no less importance, practicing psychoacoustic measures of tinnitus may become handy, as the students will evaluate real tinnitus patients in their careers. Measurements such as pitch matching are crucial in the process of specific sound therapy [31,32], and this type of practice may strengthen their confidence in their capacity to correctly evaluate tinnitus patients and set up the appropriate sound therapy if needed.

## 2. Materials and Methods

### 2.1. Participants

The simulated laboratory was part of a Master of Audiology course given during the winter semester and was mandatory for all students. Of the 22 second-year audiology students of the Université Laval program who followed the course, 21 gave their consent for the use of their laboratory results for research (1 male) (mean age = 29, SD = 7.7, range 23–44). Most of the students (*n* = 19) had normal hearing and none reported neurological disorders and external and/or middle ear pathology. A total of three students had hearing loss. One of them had mild unilateral hearing loss in the right ear, and two students had bilateral hearing loss (1 moderate, and 1 moderately severe). The two students with moderate hearing loss wore hearing aids. Four students reported having chronic tinnitus for more than six months, with three reporting bilateral tinnitus and one unilateral tinnitus (left ear). There were therefore no specific inclusion and exclusion criteria, as all of the students were required to perform the laboratory work. The data from all of the students who consented to participation in the study were analyzed, including students with hearing loss and/or pre-existing tinnitus, as they represent the real-life demographics of future audiologists. The study sample demographics are available in Table 1.

Every team of two students was asked to perform a routine audiological evaluation on each other using the soundproof room and equipment available at the university labs before the day of the simulated tinnitus lab. This evaluation included air conduction thresholds at standard frequencies (0.25 to 8 kHz per half-octave step), uncomfortable loudness levels at 0.5, 2, 4 and 8 kHz, bone conduction thresholds, word discrimination tests, and tympanometry. A case history questionnaire developed by colleagues at the Université d’Ottawa [33] was used, which included questions about socio-demographic variables (age and sex) and other audiological variables (tinnitus, self-reported hearing loss, etc.). The presence of noise sensitivity and hyperacusis were measured using standardized questionnaires, the Weinstein questionnaire [34], and the Hyperacusis questionnaire [35], respectively. As tinnitus is usually associated with reduced sound tolerance issues such as hyperacusis [36,37,38], the questionnaires were used to educate students on those issues. They had to complete the questionnaires prior to completing the simulated tinnitus lab. Instructions about the contents of the laboratory work were available two weeks before the official date of the simulated tinnitus lab. Ethical approval was obtained from the “Comité d’éthique de la recherche sectoriel en réadaptation et intégration sociale of CIUSSS de la Capitale-Nationale”, Project number #2023-2810_RIS.

### 2.2. Audiology and Psychoacoustic Measures

#### 2.2.1. Audiometry

Hearing thresholds were assessed monaurally, for both ears, by presenting pure tones from 0.25 to 8 kHz in half-octave steps using the conventional Hughson–Westlake clinical procedure [39]. Students were tested in a soundproof room using ER-3A insert earphones connected to either an AudioStar Pro (GSI) or AC-40 (interacoustics) clinical audiometers. All the audiometers were calibrated following the ANSI procedure prior to the laboratory. An otoscopy was performed before the hearing test. Hearing thresholds above 20 dB HL at any of the standard frequencies was considered to be hearing loss. Hearing loss was then classified as follows: hearing loss between 20- and 40 dB HL was considered mild, 41 to 55 dB HL moderate, 56 to 70 dB HL moderately severe, 71 and 90 dB HL severe, and hearing loss above 91 dB HL was considered profound.

#### 2.2.2. Tinnitus Pitch and Loudness Matching

The pitch and loudness matching assessments were performed in the contralateral ear from the simulated tinnitus to avoid the beating phenomenon, and were performed using the AC-40 (interacoustics) clinical audiometer only. The first step was to ask the simulated patient to describe the tinnitus in their own words. Then, the next step was to determine the timbre of the tinnitus, i.e., whether the tinnitus was more like a noise or a pure tone. The student playing the role of the tinnitus patient in the simulation was presented with three sounds individually in sequential order: a white noise, a broadband noise (with a width of one octave), and a pure tone, both of 1 kHz. The sound level of each sound was initially set at a comfortable hearing level of 30 to 40 dB above the hearing threshold (HT). Once the three sounds were presented, the student playing the role of the clinician asked the simulated patient: “which of the three sounds resembles most your tinnitus?”. Once the type of sound had been reported by the patient, the same type of sound was used for the rest of the pitch- and loudness-matching procedure. The standard clinical forced choice pitch matching procedure was used for the laboratory [29]. It consisted of presenting two sounds at two consecutive frequencies, one octave apart, starting at 1 kHz and 2 kHz. Once the two sounds were presented, the clinician asked the patient: “which of the two sounds resembles most your tinnitus?”. The clinician then increased or decreased the pitch of the two sounds presented until the same frequency was chosen twice. Once the pitch of the simulated tinnitus was determined, the same sound was used for the loudness-matching procedure. This consisted of presenting the previously pitch-matched sound at an infra threshold level and gradually increasing the volume of presentation in 2 dB steps until the patient reported just being able to hear the sound, marking the hearing threshold. Then, the volume was further increased until the patient reported that the sound matched their simulated tinnitus in terms of loudness. This procedure was repeated at least three times, and the simulated tinnitus loudness was determined by averaging the last three measured intensities.

#### 2.2.3. Minimum Masking Level and Residual Inhibition

The minimum masking level and the residual inhibition were measured using the AC-40 (interacoustics) clinical audiometer only. The minimum masking level (MML) was determined by presenting a white noise signal at an infra threshold level and gradually increasing the sound intensity in 2 dB steps until the patient reported just noticing the sound in order to measure the hearing threshold of white noise. The signal was then gradually increased in 2 dB steps until the patient reported that the white noise masked their tinnitus (MML). The procedure was repeated at least three times. Once the minimum masking level had been determined, the clinician was able to advance to the classical residual inhibition (RI) technique [40]. To determine the RI, white noise was presented at 10 dB above the MML for one minute. Right before presenting the sound stimulation, the clinician told the patient: “Now you will hear a noise for one whole minute. Following the interruption of the sound, please report any modifications that you might notice to your tinnitus”. If the student reported a change in his or her tinnitus, they were asked to inform the clinician when the tinnitus came back to a normal level. The clinician initiated a stopwatch at the offset of the one-minute stimulation and measured the time of inhibition, that is, the time required for the tinnitus to come back to a normal level. During this measurement, it was not expected that the students with the simulated tinnitus would experience RI; rather, the intent was for the student playing the clinician to understand how to carry out the classical RI assessment and request feedback on the effect of the stimulation. It is well known that residual inhibition does not occur for external sound stimulation [41], and the laboratory was not designed to simulate the temporary suppression of the sound heard by the patient in the simulation. A detailed protocol is available in Appendix A.

### 2.3. Procedure

The first student playing the role of the tinnitus patient was asked to enter the soundproof room and sit on a chair. He was fitted with one Bluetooth bone conduction headphone (HBQ-Q25, Ashata, China) on either the left or right ear (for participants with hearing loss or tinnitus, the ear tested was always the better ear). Sounds were generated using Audacity audio software. The experimenter (a clinical educator) then explained to the simulation patient that a sound mimicking the sensation of tinnitus would be presented continuously in one ear at a sensation level of about 10 dB, which is consistent with previous reports of tinnitus loudness [42,43]. The experimenter then first presented a 6 kHz tone at an audible level and decreased the volume in 1 dB steps until the student reported not hearing the sound anymore (hearing threshold). Then, the experimenter increased the level of the tone to 10 dB above the hearing threshold of the student (10 dB sensation level). A 6 kHz pure tone was chosen to mimic the real sensation of tinnitus in terms of pitch as realistically as possible [44]. A pair of Sennheiser HD 600 supra-auricular headphones was placed on the participant’s ears, and they were told that the laboratory, with their teammate acting as the audiologist, would start.

The student playing the audiologist role first asked their colleague to describe, in their own words, the tinnitus. They then performed a pitch- and loudness-matching task using the standard clinical procedure [29], in which a sound is presented in the contralateral ear, that is, the ear opposite to the one wearing the device. For the minimum masking level and the residual inhibition level, the noise masker was presented in both ears simultaneously, again following the typical standard procedure. The clinical educator was present during the whole laboratory, and could give advice to the student playing the role of the audiologist if needed. After the laboratory, the students were asked to complete an online questionnaire. This questionnaire contained three questions about their overall experience and appreciation of the laboratory, and whether they found this experience useful for their future career (see Appendix A).

### 2.4. Analysis

The analysis includes both quantitative and qualitative analysis. The written appreciation and overall experiences of the students were analyzed qualitatively. The content analysis was conducted both inductively and deductively, following the 5-phase process of qualitative data analysis [45]. First, all verbatim data were organized/transcribed into a document, and each participant was identified by a code. Then, an examiner performed a first run by sorting the data into categories relevant to our research questions. As such, certain topics, like “appreciation”, “facilitators”, “barriers”, and “suggestions”, were identified beforehand as being relevant/aligning with the purpose of our study. After the first run, the examiner conducted a second run using an inductive approach by carefully reading each written response independently and by identifying emerging ideas and sorting them into the themes that were included in the reports. Each idea or theme was identified by a code and listed in a separate file. The coding process was also iterative: if a new concept or idea emerged from the analysis of new verbatim information, previous verbatim information was re-examined. Then, an independent examiner grouped and distilled the codes into content-related categories based on overall meaning or resemblance. Major themes were identified both in terms of the number of similar codes and the number of times the code was identified by the examiner or was stated by the students. Major themes were then discussed with the first examiner to reach a non-biased consensus. Finally, the results of the content analysis were discussed and contrasted with the existing literature relevant to the focus of the study. This process was conducted to facilitate the interpretation and develop/explain our findings. For the quantitative analysis, the mean, the standard deviation, and the 95% confidence intervals were calculated for the mean group results on the psychoacoustic measures.

## 3. Results

The content analysis of the three questions in the post-laboratory questionnaire revealed three major themes: “Benefits of the laboratory on future practice”, “Barriers and facilitators of the psychoacoustic assessment”, and “Awareness of living with tinnitus”. An illustration of the results of the content analysis with examples of student reports is presented below (Figure 1).

### 3.1. Benefits of the Laboratory for Future Practice

Most of the students reported that the laboratory was useful and that they would integrate the psychoacoustic assessments of tinnitus in their future practice. When asked “Do you think these labs will help make you better future audiologists?” the entirety of the cohort responded “yes”. The major theme “Benefits of the laboratory” revealed four subthemes: “Knowledge consolidation through practical exercise”, “The reassurance of the students and future outcomes”, “Expertise in tinnitus management/support”, and “The consideration of the patients’ perspective/distress”.

#### 3.1.1. Knowledge Consolidation through Practical Exercise

This subtheme was widely reported by the students. Some students believed that it was a good means for consolidating past concepts learned in theoretical classes, as illustrated by the following two responses:


*“Without the opportunity to practice, we do not have the chance to integrate these techniques, which will necessarily be a problem when the time comes to practice them in a professional context”.*
(P01)

Many students were happy to practice not only tinnitus assessments, but also classical audiological assessment batteries carried out in clinics (audiometry, tympanometry, etc.). They found the laboratory relevant, and felt favorably towards the practical exercise, as seen in these two testimonies:


*“For the audiological assessment, I believe that we can never be “too” comfortable with the equipment, so it was also very relevant”.*
(P17)

#### 3.1.2. The Reassurance of the Students and Future Outcomes

Some students used terms such as “*rewarding*”, “*gratifying*”, or even “*fun*” (P16, 19, 02) when describing the feeling of assessing psychoacoustic measurements on their classmates. For example,

*“I found it rewarding to be able to find the frequency and sound of tinnitus and even more so if you are able to mask it”.* (P16) or *“It’s gratifying to see the person who is happy that their tinnitus stops with the white noise”.*(P19)

Additionally, some students shared their appreciation and amusement regarding the psychoacoustic measurements. For example, a student wrote:

*“I really enjoyed the experience”.* (P07) Another wrote, *“I found it fun to try to find the level of intensity and tone as close as possible to the tinnitus”.*(P02)

The students in this cohort believed that being able to practice with their classmates allowed them to prepare themselves without the pressure of being face-to-face with real patients in distress. In this sense, a student wrote:


*“I think that any practical training helps to make us better audiologists since patients will not be guinea pigs the first-time tinnitus tests are done. It also allows you to ask questions, make mistakes and exchange tips in a learning context (and without the pressure to give the best possible service to the patient). I also liked having to do the complete audiological report with a tinnitus component: it allows us to practice making complete reports and consolidate all our knowledge on tinnitus”.*
(P05)

Other students shared the belief that the laboratory prepared them to face real-life situations without the pressure induced by being faced with real patients in distress, as illustrated in the following answer:


*“It allowed me to see what a “real” assessment with a patient might look like and allowed me to get comfortable with the manipulations”.*
(P17)

Through the practical laboratory, they found reassurance and saw that they were able to overcome doubt. To illustrate this statement, two students wrote:


*“I also think it made us realize that these are accessible measures to do in the clinic, which do not necessarily take a lot of time or equipment”.*
(P04)

Finally, a student (P03) mentioned that the laboratory gave them a sense of “*empowerment*”, and that the tinnitus assessment gave them insight into the possibility of doing something for these patients. This shows that a practical tinnitus lab can go a long way in reassuring students of the skillset they have acquired during their academic pathway and their overall future professional utility.

#### 3.1.3. Expertise in Tinnitus Management/Support

Some students placed an emphasis on tinnitus management, and how future tinnitus patients will benefit from what they learned during the course. To illustrate this statement, a student wrote:


*“Knowing how to perform this procedure can be beneficial in the management of tinnitus patients. This lab has added a tool to our audiologist toolbox”.*
(P01)


*“This laboratory allowed us to have more knowledge and concrete expertise on the possibilities and tools we have as clinicians to help our tinnitus patients”.*
(P16)

Different aspects of the management of patients with tinnitus were mentioned, including coping strategies for tinnitus distress through counseling. The next three sentences illustrate this point.


*“I find it very relevant to know how to assess the frequency and intensity of tinnitus to improve our counselling with patients. Also, the demonstration of tinnitus masking could reduce the distress experienced by offering a temporary solution for tinnitus management”.*
(P18)


*“Also, we were able to experience a tiny part of what people experience with tinnitus. This allows us to be more empathetic towards them in the future”.*
(P20)

Some students were even able to link their clinical assessment with therapeutic gateways and patient referral/guidance. As such, a student reported:


*“This will allow us to better focus our questions when a patient presents with disturbing tinnitus. By playing the role of the patient, it also allows us to prepare ourselves in our intervention by better guiding the patient and reassuring him about the test”.*
(P13)

A student mentioned that there is currently a flaw in the health system regarding treating and taking care of the population experiencing tinnitus, and that these laboratories gave them a certain “expertise” in tinnitus management. They explained:

*“They* [the laboratories] *will bring expertise to the labour market and try to find solutions where the current service contains several gaps”.*
(P03)

#### 3.1.4. The Consideration of the Patients’ Perspective/Distress

A fair number of students (*n* = 5) pointed out the usefulness of the tinnitus assessment protocol for managing patient distress. They believed that these measures could come in handy when a patient exhibiting distress consults them in the future. In this sense, two students stated the following:


*“I am sure that performing these tests with tinnitus clients would allow some individuals to feel understood, listened to and supported in their efforts. Having a hearing health professional try to objectify their tinnitus could improve trust and, to some extent, create hope and motivation in finding solutions”.*
(P05)

Other students linked the tinnitus assessment protocol to the management of tinnitus distress, as shown in the following statement:


*“Also, the demonstration of tinnitus masking could reduce the distress experienced by offering a temporary solution for tinnitus management”.*
(P18)

Interestingly, the cohort seems to have learned the implications of their role as an audiologist in the treatment of tinnitus, and how they could use this laboratory to help tinnitus patients to cope with their symptoms and earn their trust from a therapeutic relationship perspective. Two quotations of two students illustrate this:


*“These are tests that allow us to show the customer that we take his complaint into account, that it is taken seriously and that he is not “crazy”. This shows our client that we have tried to understand his problem and that we are trying to find a solution. I believe that this promotes a better therapeutic relationship with our client and increases the client’s trust in us”.*
(P08)

These statements show how the students are already inferring the multiple uses these measures could have in their future practice (referring, reassuring, selling argument, and so on). A suitable number of students reported “patient-oriented benefits of assessing psychoacoustic measures of tinnitus”. Some students reported that assessing these measures grants the clinician the ability to “objectify a subjective experience”, as stated in the following transcript:


*“Since tinnitus is a subjective experience experienced by the patient, I found it interesting that we can somehow “objectify” his tinnitus”.*
(P04)

### 3.2. Barriers and Facilitators

While most students found the tinnitus assessment battery to be relatively easy to complete, a certain number of students encountered difficulties in successfully passing the assessment. Thus, another significant theme commonly reported by the students was the presence of barriers and facilitators. In this major theme, three subthemes were identified: “The clearness of the protocol”, “Difficulties and improvements”, and “The impact and consideration of past and future internships”.

#### 3.2.1. Clearness of the Protocol

Many students recognized the significance of employing clear and simple language that is easily comprehensible to facilitate the process of identifying the patient’s tinnitus sensation. To illustrate this subtheme, one student expressed the following:


*“It was difficult for me to give clear instructions, the wording was clearly to be reworked, especially to give instructions to someone who does not have knowledge in the field of audiology”.*
(P03)

However, some students appreciated the way in which the procedure was presented to them. The verbatim reports indicate the simplicity of the different steps described in the written procedure used. As such, one student reported:

*“The steps to follow were very clear”.* (P02) Another student reported *“Being the clinician wasn’t too difficult because we were following a clear protocol, and it was easy to understand what I was doing as well”.*
(P20)

A student mentioned “non-verbal cues” as a solution used to enrich their assessment, as illustrated in the following statement:


*“It is our duty to monitor the patient’s non-verbal to adapt our methodology”.*
(P05)

#### 3.2.2. Difficulties and Improvements

A barrier certain students had to face during the laboratory was technical difficulties and the presence of real tinnitus. Some students reported the lack of familiarity with the audiometers at their disposal, causing them to lose precious time during the laboratory session. To illustrate this, a student reported not feeling competent enough to carry on the assessment, stating:

*“Since this was my first-time doing tinnitus psychoacoustic measurements, I felt incompetent”.* (P06) They explained, *“I do not fully master the operation of the audiometer. I am still looking for myself (when I have to use channel 2, for example)”.*
(P06)

On the other hand, certain students reported the psychoacoustic measurements performed during the laboratory to be straight to the point and easy. To illustrate this subtheme, two different students made the following statements:


*“I was surprised by the steps that are not very long to perform”.*
(P12)


*“The procedures were easier and more instinctive than I imagined”.*
(P15)

This goes to show that there is a lot of variability in the perceived competence of the students. As reported in the demographics table, four students had chronic tinnitus. When reporting their experience as the simulated tinnitus patient, three out of four noted difficulties related to the interaction of the task with their real sensation of tinnitus. They found their tinnitus to represent an added difficulty to the task, as they had to divide their attention between the two perceptions. As such, a student wrote:


*“I already had tinnitus on the right and there I also had tinnitus on the left. So it was harder for concentration. It was sometimes difficult for me to tell the difference between the two”.*
(P09)

Another student wrote:


*“I think my answers were skewed because my attention was confused between the real tinnitus coming from the other ear”.*
(P01)

Although the effect of having real tinnitus on the difficulty of the task could have been anticipated, one of the students used her tinnitus to her advantage during the assessment. She noted:


*“I had to use my tinnitus to help me spot the one that was stimulated. Even though I wasn’t aware it was there, I noticed the silence when I removed the fake tinnitus”.*
(P14)

Lastly, two students made some suggestions to improve the laboratory. One believed that including otoacoustic emission assessments in the current lab would have been relevant, while the other would have wanted to try different narrow-band noise frequencies to seek better RI effects. Both students placed an emphasis on “More opportunity to practice”, as stated by the two students in the following:


*“It would also be very beneficial to have access to the audiology laboratory to be able to practice the procedures that are taught. Without the opportunity to practice, we do not have the chance to integrate these techniques, which will necessarily be a problem when the time comes to practice them in a professional context”.*
(P01)


*“I would have liked to have done it again more than once, with other frequencies. More practice to try to make the residual inhibition work”.*
(P11)

A student also suggested that a step-by-step explanatory video explaining the tinnitus assessment could be viewed beforehand to ensure a good understanding of the protocol as a whole. They reported:


*“To achieve an optimal simulation and an optimal learning experience, I could have benefited from a little more explanation beforehand. An explanatory and demonstrative video such as those offered in the Hearing Aids 1,2,3 courses could be an asset”.*
(P01)

#### 3.2.3. The Impact and Consideration of Past and Future Internships

Some students were more prepared than others for the laboratory, as some did similar tinnitus assessments during previous internships. To illustrate this, a student reported:


*“I felt in control because I arrived well prepared at this laboratory, and I had already observed my (clinical) supervisor perform this kind of assessment”.*
(P10)

Others reported how the psychoacoustic lab “reflected” what they saw during previous internships. As such, a student reported:


*“This was exactly what I had done at the IRDPQ [Quebec rehabilitation center] during my internship: to give the questionnaires on tinnitus and sound hypersensitivity to patients at first, and then to match their tinnitus in frequency and sound (with residual inhibition). This laboratory is therefore useful and will perfectly reflect the practice of audiology, for those who will work in rehabilitation with tinnitus patients”.*
(P19)

Finally, some mentioned how the psychoacoustic tinnitus laboratory was completely new, and was absent from internships undertaken in the past. As stated here:


*“It is rare in our internships to be able to experience a tinnitus evaluation, so being able to do so and be supervised helps us for when we meet tinnitus patients”.*
(P20)

On a further note, some students mentioned how the labs prepared them for future internships, as mentioned here:


*“The first laboratory allows us to put into practice our learnings of the past year and to prepare for our next internships”.*
(P05)

This goes to show how diverse the directions of the learning process (the state of knowledge/skills) of students are, and how the described laboratory unifies the understanding of the students by giving them a standardized skillset and tools to overcome situations they will face in the future.

### 3.3. Awareness of Living with Tinnitus

The content analysis of the post-laboratory questionnaire revealed a third and final main theme: “Awareness of living with chronic tinnitus”. This theme was subcategorized into three subthemes: “The sensation of tinnitus”, “The subjectivity of tinnitus experience” and “Being tested for tinnitus”.

#### 3.3.1. The Sensation of Tinnitus

Many students got a glimpse for the first time of what it is like to have the persistent sensation of tinnitus. This led the students to report what came to their minds while experiencing the 6 kHz pure-tone stimulation. The words used to describe their sensation varied from one student to another and included the words “*disturbing*” (P12, P07), “*uncomfortable*” (P09), “*annoying*” (P20) and even “*invasive*” (P07) and “*strange*” (P11). These reports resemble what is commonly reported by real tinnitus patients and their overall experience with their tinnitus sensations [12].

#### 3.3.2. The Subjectivity of Tinnitus

A fair number of students shared that experiencing tinnitus for the first time was an eye-opening experience and helped them better understand the difficulties faced by this population in their daily lives, while also performing the test. The laboratory seems to have had an impact on the students by raising their awareness of the lived experience of tinnitus patients. The following two quotations by two students illustrate this notion:


*“It’s hard to try to objectify a subjective experience, so I’m aware that it must also be difficult for a patient to answer the different tests”.*
(P15)


*“I can therefore understand that despite the tests carried out, the measurements obtained are not completely “valid”, since they are very subjective data dependent on the unique experience of the patient”.*
(P05)

Many students found it difficult to be tested as tinnitus patients. When they were placed in the soundproof room and isolated while hearing the simulated tinnitus sound, most students reported having a hard time answering the clinician’s questions, as they found it difficult to compare and match their simulated tinnitus with the sound stimulations presented by their classmates. To illustrate this, a student wrote:


*“As it is a very subjective phenomenon, it is sometimes difficult to answer and above all, to give the answer expected by the clinician”.*
(P03)

Another student wrote:


*“By playing the role of the simulated patient, one comes to better understand the difficulty of the task as well as the uncertainty of the answers given. It was difficult to compare the intensity and frequency of our tinnitus to the sound sent by the audiologist”.*
(P13)

#### 3.3.3. Being Tested for Tinnitus

Several students reported difficulties during the task when playing the role of the simulated tinnitus patient. As such, a student reported:


*“We needed a lot of concentration and also try to be as honest as possible about our answer, which was not always easy”.*
(P13)

As mentioned by the student above, “concentration” was often reported as a key factor induced by the tinnitus assessment. Many students had difficulties focusing their attention on the task. They believed the task was very demanding in this regard, as stated here:


*“I found it rather difficult to pay attention to tinnitus and be able to recognize how often and how similar it sounded”.*
(P04)

Some students reported being “tired” by the experience, as illustrated in the following statement:


*“I found the experience tiring, especially the stage where there was the addition of white noise”.*
(P10)

Attention, concentration, and fatigue were widely shared as factors that impacted the difficulty of simulated tinnitus patient task. Some students reported they had a hard time during the task because the sounds presented by the clinician distracted them from hearing their tinnitus. The phenomenon of “habituation” was reported, though none of the students had been exposed to this concept during previous theoretical classes. To illustrate this, a student wrote:


*“A few seconds after the manager started the sound, it faded. I felt like my brain was quickly judging that the sound was irrelevant and making sure I couldn’t hear it anymore”.*
(P04)

Another student wrote:


*“Over the course of the experiment, by hearing several sounds, we come to no longer hear our simulated tinnitus, which makes it difficult to give reproducible answers to the clinician”.*
(P21)

To sum up the general opinion and overall experience shared by the cohort with respect to the laboratory on tinnitus assessment, a student wrote:


*“These two labs will make me a better audiologist because they allowed me to experiment tinnitus assessment from A to Z through interviews, questionnaires, audiometry and psychoacoustic measurements. I think it’s important to have tried experimentation. Having had the chance to experience both sides of the coin (clinician/patient) is a plus because it allows us to put ourselves in the patient’s shoes. Thus, I will be able to understand that it is not an easy task to measure the frequency and sound of tinnitus and that it is tiring”.*
(P10)

### 3.4. Laboratory Psychoacoustic Results

In response to the first question regarding the general form and description of the simulated 6 kHz pure tone stimulation, the entirety of the class stated that it was a continuous, high-pitched hissing or whistling sound. In response to the second question regarding the general form of the tinnitus sound in comparison to the three sounds presented (white noise, narrowband noise, pure tone), all of the students stated that the simulated tinnitus most resembled the pure tone stimulation. The results of the psychoacoustic assessment task are presented in Table 2. For the pitch-matching task, the mean frequency of the sample was 6727 Hz, 95% CI [6466, 6987]. For the loudness-matching task, the mean loudness of the sample was 9 dB SL, 95% CI [7.97, 10.12]. The confidence intervals of the pitch- and loudness-matching tasks confirm that the averages of the group were not different from the targets, which were 6 kHz and 10 dB SL, respectively.

## 4. Discussion

The primary objective of this research project was to determine whether a tinnitus simulation method could serve as an effective means for students to practice tinnitus assessment measures, while also providing them with a deeper understanding of what it means to experience tinnitus and undergo a clinical evaluation by an audiologist. This research was carried out among a group of second-year students enrolled in the audiology program at Université Laval, as an integral component of their program’s curriculum. Consequently, all students that gave their consent for the use of their laboratory results and experiences were included in the study. During this laboratory, students were asked to take turns playing the clinician and the patient and to assess a simulated tinnitus with a fixed pitch and loudness. The results of this study showed that, in general, students reported multiple benefits of the practical laboratory, including knowledge consolidation through practical exercise, reassurance in their capacity to complete the assessments without being confronted with real-life patients in distress, while also providing them with a certain “expertise” in tinnitus management. The students also reported being confronted with different barriers and facilitators during the practical lab, which included the significance of having a well-defined protocol and giving precise instructions, along with technical challenges audiologists must learn to overcome in their clinical practice and the influence of previous internships and other experiences on their overall confidence and comprehension of the procedure. Finally, students were given the opportunity to gain a glimpse into the actual experience the real sensation of tinnitus for the first time (for most students), and contemplate the challenges that may arise when feeling annoyed or distressed by the sensation. They shared their thoughts on the physical sensation of the simulated tinnitus, and reflected on the subjectivity of tinnitus and how it made them “aware that it must be difficult for a patient to answer the different tests” (P15). Additionally, they reported various challenges that may arise when being tested for tinnitus, including concentration, fatigue, and making sense of the different sounds that are presented by the audiologist. Lastly, the students in the cohort demonstrated a satisfactory ability to evaluate the simulated tinnitus, despite an occasional lack of confidence, when completing the tinnitus assessment battery.

In Section 4.1, we explore various clinical simulations used in the field of audiology and health care, highlighting their advantages as instructional techniques, and the ways in which we believe our method can address the current underuse of simulations in the specific area of tinnitus testing and management. In Section 4.2, we discuss the overall performance of the students in assessing the simulated tinnitus used in our technique, and reflect on if and how we should evaluate the performance of each student based on the results of their psychoacoustic assessments. Finally, we discuss the extensive use of psychoacoustic measures of tinnitus in tinnitus research, making a contribution not only to our understanding of the underlying mechanisms of tinnitus, but also serving as a valuable tool for guiding clinical practice and facilitating patient counseling, thus highlighting the crucial significance of the mastery of this technique by students before facing real patients in distress.

### 4.1. Clinical Simulation in Audiology

Clinical simulation can be used for both evaluating clinical competencies and/or as an experiential learning opportunity for students [23]. Five categories of health care simulation, varying in terms of how closely the simulation replicates the real-world experience with respect to physical, environmental and psychological elements, have been proposed by Lopreiato and collaborators (2016) [46]. These five categories include (1) standardized patients, (2) (part) task trainers, (3) mannequins, (4) computer-based simulations, and (5) immersive virtual reality. Briefly, standardized patients involve training someone to act while simulating an actual patient in a realistic and standardized way, (part) task trainers usually consist of a device to train a specific procedure or skill, mannequins consist of a life-sized human-like simulator, computer-based simulations consist of a simulation represented on a computer screen, and finally, immersive virtual reality is a computer-based three-dimensional representation [23].

Clinical simulation is not widely used by audiology and speech–language pathology educators, as reported in a recent study [23]. From the 136 respondents to their survey, they found that only 51% reported using simulations for clinical education. Most of them reported using standardized patients and/or computer-based simulations. They also noted different barriers to the use of this method, such as a lack of knowledge/training, limited financial resources, an under-trained faculty, and minimal guidance from accrediting bodies. Clinical simulation has been used in audiology for many diverse purposes, such as improving (1) case history and feedbacks skills [47], (2) audiometry testing skills (including visual reinforcement audiometry (VRA)) [48,49], (3) auditory brain-stem response setup skills [50] and waveforms analysis [51], (4) otoscopic examination [52], and (5) probe tube placement for hearing aid verification [53]. The latter uses a real-sized head and ear mannequin called CARL that allows the many different audiology skill sets to be practiced, including ear impression, ear wax removal, audiometry, and hearing aid fitting/verification [54].

To the best of our knowledge, the simulation of tinnitus for teaching purposes has never been reported in the literature. The CARL system described above describes the possibility of verifying tinnitus maskers, but it seems to be the only report of tinnitus in the clinical simulation field [54]. This is not surprising, considering that tinnitus evaluation and management has been incorporated into the standard requirements only recently (August 2017). The limited hours devoted to tinnitus within the audiology practicum may also explain the limited development of teaching methods. As mentioned by Henry et al. (2021), in a study on the tinnitus training of graduate audiology students, only 10 out of 32 US programs devoted one credit hour (sixteen total hours of class and clinic time) to tinnitus. Additionally, clinical placement with tinnitus-specific training and mentorship has also been regarded as problematic, with only 41% of students receiving such training and mentorship. As stated by Dr Fagelson in a recent opinion paper, it may not be reasonable to expect students to embrace tinnitus management in their careers if they do not receive a critical mass of experience and information supporting tinnitus management during their matriculation through the audiology program [55].

We believe that the method developed here may improve the situation. First, the technique is relatively simple and cheap, only requiring a bone transducer and standard audiology equipment (audiometer, soundproof room, etc.), to which programs are accustomed. The method incorporates almost all the elements needed in order to create an effective learning environment as described by So et al. (2019): (1) a situation that they would encounter under normal circumstances (psychoacoustic measurements of tinnitus can be performed in clinics); (2) an environment resembling a real clinical setting (the simulation was performed in a room that is similar to an audiology room); (3) equipment that they would use in real practice (they used a clinical audiometer and typical headphones of the sort they would use in clinics); (4) the learning experience is problem centered and is similar to real clinical encounters (assessing a tinnitus patient and having them perform a psychoacoustic test is similar to a real clinical encounter); (5) learners need to feel safe to express themselves (students were accompanied by a clinical educator who assisted them during the laboratory, some students reported that this was an asset); (6) learners receive timely feedback from different sources (students received feedback from the clinical educator). Playing the role of the audiologist while testing a colleague, rather than a patient, provides an environment that is conducive to learning: the student can ask questions of the clinical educator and can make mistakes without the pressure of providing services to a real patient.

This method can be regarded as a first step before seeing actual tinnitus patients. As mentioned by Fagelson (2023), a lot of tinnitus patients experience co-occurring mental health disorders, and these can be hard to handle for audiology interns, especially when they have no previous experience with the evaluation and management of tinnitus. As reported here by some students, the laboratory substantially enhanced their confidence in tinnitus management. The laboratory offers the opportunity for each student to manipulate the equipment and test a simulated patient. Inviting real tinnitus patients to be tested at the university laboratory for teaching purposes would not be feasible, as it would be too costly. Most of the benefits discussed above are related to when the students play the role of the clinician, but we believe that playing the patient role is of no less importance. Indeed, being tested for tinnitus while hearing a constant high-pitched sound for 30 to 45 min gave the students the opportunity to experience tinnitus and being tested for tinnitus. It can be seen from the themes that emerge from the qualitative analysis that many students experienced annoyance at hearing the sound constantly, and difficulty in describing this very subjective phenomenon, and finally the difficulty of performing the psychoacoustic tasks from the perspective of the person being tested. As stated in a scoping review of healthcare professionals learning from the simulation of the experience of patient illnesses [28], ‘point-of-view’ simulations in healthcare education may positively promote attitudes towards empathic care and a desire to be more understanding and diligent and to demonstrate kindness toward patients in general. Other benefits of these techniques include the generation of an increased confidence in teaching and performing self-management skills, recognition of the challenges and awareness of possibilities for improvement, feelings of empowerment and ownership, and a reduction in the negative perceptions and stigma associated with the illness. We believe that this experiential learning opportunity will be of great value for students, and will help them manage future tinnitus patients. However, we do acknowledge that the student playing the role of the tinnitus patient is not a real patient, which is a limitation. Students’ exposure to real patients should be included during internships or laboratories devoted to evaluating real patients.

### 4.2. Student Performance

The performance of the students regarding the application of the protocol and the results obtained during the laboratory were not assessed in the current study per se, meaning that, although the students were asked to note the results for each measurement, they were not evaluated on the accuracy of their measurements. No grade was assigned to their performance. They were also informed, prior to the laboratory, that they would not be evaluated on how well they performed the measurements or on the exactitude of their results with respect to the simulated tinnitus. That being said, by knowing the characteristics of the target sound/simulated tinnitus, the accuracy of the group of audiology students when performing the tinnitus pitch and loudness matching can be assessed. Indeed, as reported in Table 2, the average results for pitch and loudness matching were 6727 Hz and 9 dB SL, respectively, which is very accurate, considering that the tinnitus pitch was 6000 Hz, and was set at 10 dB SL for all simulated tinnitus patients. In addition, these values were included in the 95% confidence intervals of the students’ results, suggesting that there was no difference between the target and the mean group result. The lack of confidence expressed by many students regarding their aptitude in performing these measurements is not supported by these results; they are, overall, accurate. We do not believe that assessing how close the tinnitus pitch and loudness matching results were to the target would be appropriate as an evaluation method. Indeed, the results of the measurements are not the responsibility of the student performing the measurement alone, but also the student playing the patient. If, for some reason, the latter is not responding properly, the pitch and loudness matching may be far away from the target/simulated tinnitus. This could penalize the student performing the measurements, despite their proper application of the method. However, we believe that assessing the applicability of psychoacoustic measurements on the basis of a laboratory test would be an effective evaluation method. Indeed, the educator could evaluate how well the student follows the procedure and explains the measurements to the patient, and whether they use the equipment adequately. This constitutes an evaluation of the process rather than the results. To move forward with evaluating their applicability, students should have the possibility of practicing performing the measurements more than once. Many students mentioned their desire for more practice of these techniques in order to foster their confidence and level up their skills.

On a further note, 19 out of 21 students did not report any change in the sensation of the simulated tinnitus following the offset of the white noise stimulation during the RI assessment, which was expected, given what is known about the RI of external sounds [41]. Still, two students reported experiencing RI for up to 16 s after the offset of the sound (cf. Table 2). We see only two likely explanations for those reports: one is attention and the other is response bias. For attention, the two students may have been distracted or not paying attention to their simulated tinnitus when the white noise presentation stopped, thus explaining the duration of inhibition. It is also plausible that the two students gave an answer to please the teammate playing the clinician. The laboratory could be improved by mimicking residual inhibition: it would be possible to recreate the experience of the RI of tinnitus by stopping the sound presentation and then subsequently increasing the intensity of the simulated tinnitus back to a normal level after a few seconds.

### 4.3. Psychoacoustic Measurements of Tinnitus: Underestimated Measurements?

For more than eight decades, now, psychoacoustic measurements have been used to better understand the pathophysiology of tinnitus and to guide clinical interventions [56,57]. Studies investigating tinnitus pitch have shown that the tinnitus sound is usually composed of the same frequencies as those affected by hearing loss [9,10,58], leading to the idea that the neuronal signal responsible for tinnitus may arise from maladaptive neuroplasticity mechanisms within the central auditory system following hearing loss [59,60,61]. This type of tinnitus is called central tinnitus. Tinnitus for which the aberrant neuronal signal is generated at the periphery is called peripheral tinnitus [62]. It is possible for the two forms to co-exist within an individual. Recent studies on tinnitus masking and residual inhibition have shown that it may be possible to differentiate these two subtypes [41,63]. On the clinical side, these measurements have long been used as predictors for the success of different sound therapies including tinnitus maskers [40] and hearing aid amplification [64]. Briefly, the chances of success of these sound therapies increase when the tinnitus is easily maskable [64] and/or when the frequency region including the tinnitus pitch is stimulated [65,66]. Psychoacoustic measurements of tinnitus such as pitch matching are also warranted in the implementation of new tinnitus therapies [67]. For example, tailor-made notched-music therapy consists of listening to a piece of music in which the frequency of the tinnitus has been filtered out [68,69]. It thus requires measuring the tinnitus pitch to individualize the therapy to the patient’s tinnitus. The individualization/personalization of tinnitus therapy on the basis of psychoacoustic measurements, including sound-based, bimodal and neuromodulation therapies, is increasing [32]. Needless to say, if audiology as a profession wants to become the go-to provider of such therapies, these measurements need to be mastered. Additionally, tinnitus psychoacoustic measurements are also considered by ASHA and AAA to be an important component of tinnitus counseling [70].

A lot of students reported that the laboratory had improved their tinnitus counseling skills, both in terms of what the measurements can provide to a patient (temporary relief, a better understanding of their tinnitus, etc.) and their experience of being the patient (better understanding of what it feels like to be tested for tinnitus and living with tinnitus). These findings are consistent with what is known on the short-term effects of the simulation of the experience of illness on learners [28]. Still, psychoacoustic measures of tinnitus are not widely used in clinical settings by audiologists in the US and Canada [19]. This may not be surprising, considering that specific training devoted to tinnitus was only incorporated into the audiology curriculum very recently in the US [20]. In addition, the 16 total hours of class and clinical time on average that are devoted to tinnitus may not be sufficient to include these measurements. It is not known whether psychoacoustic measures of tinnitus are taught at school and/or during clinical practica, or, if so, how they are taught. This may include theoretical knowledge and/or practical knowledge about these measurements. A few students in the current study mentioned that they did not have the opportunity to practice performing those measurements during their internships. This may be the case for many audiology students around the world. The tinnitus simulation laboratory developed here offers a unique opportunity to move from theory to practice, and allows students to experience something akin to tinnitus. Moreover, the laboratory provides the basic practical knowledge required to implement the four basic psychoacoustic measurements. These might come in handy for delivering individualized therapies now and in the future. As mentioned by some students, performing these measurements will strengthen their confidence in being able to conduct tinnitus assessments, provide counseling, and perform tinnitus interventions in their future careers. The lived experience of both the audiologist evaluating the tinnitus patient and the tinnitus patient being evaluated will likely improve all aspects of the students’ future practice. Therefore, this laboratory meets all of the standards of the tinnitus accreditation training program, improving assessment, counseling, and intervention.

### 4.4. Limitations and Perspectives

The qualitative content analysis carried out in our study followed the five-phase process of qualitative data analysis to limit any known bias that might arise in this type of study [45]. Two independent examiners were brought together to extract and make sense of the verbatim statements reported by the study sample. Students were notified beforehand that their data would not be analyzed by their professor and that their responses would have no impact on the marks associated with the course in which the psychoacoustic laboratory took place. Still, we cannot exclude the possibility that some students were biased in thinking that giving positive feedback on their experience would somehow improve their grade. Additionally, this study was carried out on a single cohort of audiology students within a single audiology program of the Université Laval in Québec, Canada. More students from different programs should be tested to confirm the findings of the present study. Bearing this in mind, one of the purposes of this study was to make accessible to the scientific and academic community the evaluation sheets and methods associated with tinnitus psychoacoustic assessment training. From the data of the current study, it is not possible to quantify the impact of the laboratory on the students’ knowledge and understanding of the psychoacoustic measures of tinnitus, specifically, or of tinnitus more generally. A follow-up study could compare the final scores of those who performed the laboratory with those who did not among equivalent cohorts of students.

As discussed above, certain adjustments to the procedure could be made to improve the RI simulation by stopping the sound presentation and then subsequently increasing the intensity of the simulated tinnitus back to a normal level after a few seconds. It is also admitted throughout this study that the psychoacoustic evaluation carried out by the students is not the only type of evaluation that could/should be performed by audiologists when confronted with tinnitus patients. As such, the laboratory could benefit from the implementation of a full report with (but not limited to) the administration of various questionnaires related to the impact of tinnitus on the patients’ quality of life (such as the THI or TFI). Additionally, the students were asked to wear the device producing the tinnitus sound for between 30 and 45 min while sitting in an audio booth. This short duration of the simulated experience of tinnitus may not have been long enough for them to experience the full extent of the impact of tinnitus on all aspects of a patient’s life. Moreover, they did not experience the simulated tinnitus in various contexts and conditions, such as when talking to a friend in the cafeteria or when trying to fall asleep in their bed. Increasing the duration of this experience, such as for a whole day, and having them wear the device in various situations may increase their understanding of what is like to live with tinnitus. Finally, as tinnitus is usually associated with some level of hearing loss, it could be possible to simulate unilateral hearing loss, with a foam earplug for instance, in the ear where the simulation takes place to better illustrate typical profiles of patients with unilateral tone-type tinnitus.

## 5. Conclusions

Student audiology training in tinnitus evaluation and management is heterogeneous, and has been found to be insufficient. To encourage better training in this field, we designed a new clinical simulation laboratory for tinnitus evaluation: one student plays the role of the tinnitus patient, wearing a device on one ear that plays a sound like tinnitus, while another student plays the role of the audiologist evaluating their condition. This simulation technique allows the student playing the audiologist to practice the standard psychoacoustic measurements of tinnitus in a similar fashion to that of testing a real tinnitus patient. From the students’ perspective, this method allowed them to practice performing these measurements in a controlled environment with the counseling of educators and without the pressure of facing a real patient. In addition, being the patient allowed them to experience tinnitus and being tested for tinnitus. They reported that this laboratory would have a tremendous positive impact on their ability to manage tinnitus patients in their future career. Although this technique seemed to have a positive impact on the students’ confidence in their abilities to carry out the psychoacoustic assessment, as well as to guide and care for future patients with tinnitus, the long-term impacts on the students remain unknown. We hope that this cheap, fast, and effective simulation laboratory will be used by audiology educators and other healthcare educators for the training of students in tinnitus evaluation and management.

## Figures and Tables

**Figure 1 brainsci-13-01338-f001:**
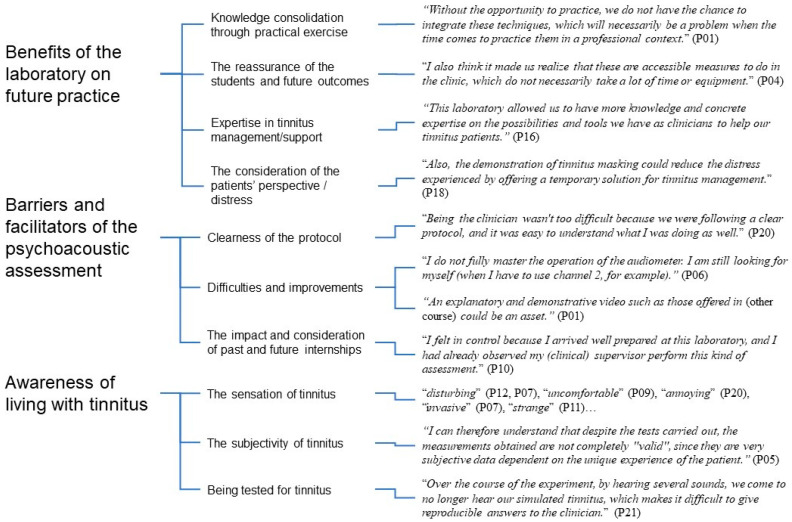
Illustration of the results of the qualitative content analysis of the post laboratory questionnaire with examples of student reports.

**Table 1 brainsci-13-01338-t001:** Sociodemographic characteristics of the participants.

Participant Number	Age (Yrs)	Sex (F/M)	Chronic Tinnitus (Y/N)	Simulated Tinnitus Laterality (R/L)	Audiometric Profile (Standard Frequencies, 0.25–8 kHz)
P01	37	M	Y	R	NH
P02	41	F	Y	L	NH sloping to a mild unilateral right HL
P03	24	F	N	R	NH
P04	25	F	N	L	Mild bilateral HL
P05	24	F	Y	L	NH
P06	44	F	Y	L	Mild bilateral HL sloping to moderately severe
P07	23	F	N	L	NH
P08	25	F	N	L	NH
P09	36	F	N	L	NH
P10	42	F	N	L	NH
P11	24	F	N	R	NH
P12	24	F	N	L	NH
P13	23	F	N	R	NH
P14	25	F	N	L	NH
P15	26	F	N	R	NH
P16	28	F	N	R	NH
P17	26	F	N	L	NH
P18	27	F	N	L	NH
P19	27	F	N	L	NH
P20	26	F	N	R	NH
P21	29	F	N	R	NH

Yrs = years, Y = yes, N = No, F = Female, M = Male, R = Right, L = Left, NH = Normal Hearing, HL = Hearing loss.

**Table 2 brainsci-13-01338-t002:** Results of the different psychoacoustic measurements of the group.

Type of Measure	Mean	SD	Min	Max
Pitch matching (Hz)	6727	1285	4000	8000
HT at tinnitus pitch (dB HL)	9	16	−5	65
Loudness matching (dB SL)	9	5	2	20
HT with white noise (dB HL)	19	15	5	60
Minimum masking level (dB SL)	23	8	6	36
	Positive/Negative			
Residual inhibition	2 */20			

* The durations of the residual inhibition, as measured by the students, were 14 s and 16 s.

## Data Availability

The data sets generated and/or analyzed during this study are available from the corresponding author upon reasonable request.

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
