# Peer review of "Improving Audiology Student Training by Clinical Simulation of Tinnitus: A Glimpse of the Lived Experience of Tinnitus"

_brainsci, 2023, doi:10.3390/brainsci13091338_

Round 1
Reviewer 1 Report
The aim of the study is to explore the benefit of a new simulated tinnitus patient approach in teaching audiology students the psychoacoustic assessment of tinnitus.The proposal of clinical simulation of tinnitus is interesting.
The Tinnitus Psychoacoustics Laboratory Evaluation Sheet and the post laboratory questionnaire are appropriate and complete.
Line 134. Material and methods: could you use the term chronic instead continuous tinnitus (as reported in the table 1)?
Pag. 6-11. Results: the statements of the students are redundant, I suggest you to synthesize them.
I also suggest to comment in the introtuction and in the discussion a limitation of the study:
Psychoacoustic measures of tinnitus are relevant but not sufficient in the tinnitus evaluation, based on international tinnitus guidelines. The figure of audiologist is significant but a multidisciplinary approach is widely supported for tinnitus patients. The key role of questionnaires (THI, TFI, TQ, BDI, BAI, HADS, ecc) is not mentioned in the manuscript. The gold standard treatment for chronic tinnitus is cognitive behavioural therapy (CBT) and decompensated tinnitus patients may benefit of CBT alone or with sound therapies.
Author Response
Reviewer 1:
The aim of the study is to explore the benefit of a new simulated tinnitus patient approach in teaching audiology students the psychoacoustic assessment of tinnitus. The proposal of clinical simulation of tinnitus is interesting.
The Tinnitus Psychoacoustics Laboratory Evaluation Sheet and the post laboratory questionnaire are appropriate and complete.
Line 134. Material and methods: could you use the term chronic instead of continuous tinnitus (as reported in table 1)?
Reply: As suggested, the term continuous tinnitus was changed to chronic tinnitus in the Materials and methods section and elsewhere, when appropriate.
Pag. 6-11. Results: the statements of the students are redundant; I suggest you synthesize them.
Reply: We agree that some of the student reports are redundant. We removed some of the reports (P16, line 386, P15 line 395, P21 line 420, P05 line 448, P09 line 463, P12 line 476, P04 line 531, P02 line 547, P17 line 680) and also provided a new illustration that synthesizes the results of the qualitative analysis (see Figure 1 below). This figure was added at the beginning of the results section, line 353. We believe that both, the figure, and the removal of some reports, improved the readability of the manuscript. We thank the reviewer for his comment.
Figure 1. Illustration of the results of the qualitative content analysis of the post laboratory questionnaire with examples of student reports.
I also suggest to comment in the introduction and in the discussion a limitation of the study: Psychoacoustic measures of tinnitus are relevant but not sufficient in the tinnitus evaluation, based on international tinnitus guidelines. The figure of audiologist is significant but a multidisciplinary approach is widely supported for tinnitus patients.
Reply: We thank the reviewer for this comment. We have now added in 59: « The role of an audiologist is significant in tinnitus evaluation and management but should be integrated within a multidisciplinary approach that is widely preferred and supported for tinnitus patients including the involvement of various healthcare professionals such as psychologists, otolaryngologists, psychiatrists and physiotherapists, to name a few [18] »
The key role of questionnaires (THI, TFI, TQ, BDI, BAI, HADS, etc) is not mentioned in the manuscript.
Reply: We thank the reviewer for this comment. We have now added in 59: « The impact of tinnitus on quality of life is usually assessed through the use of various standardized questionnaires such as the tinnitus handicap Inventory [13] and the tinnitus functional index [14]. » This was also added in the limitations/perspective of the study at the end of the manuscript.
The gold standard treatment for chronic tinnitus is cognitive behavioural therapy (CBT) and decompensated tinnitus patients may benefit of CBT alone or with sound therapies.
Reply: We thank the reviewer for this comment. We have now added in 60: « Currently, cognitive behavioural therapy has been shown to be the most effective treatment for tinnitus [15].»
Reviewer 2 Report
I would like to thank the authors for their submission and allowing me to review their work.
This is an interesting study on an important topic. However, I would be grateful if you could add further explanations and changes on the following points:
1) ABSTRACT: Page 1, line 19
Please specify the mean age (± standard deviation) and gender of the study population.
2) ABSTRACT: Page 1, line 21
Where and when was the study conducted? Please specify in the methods section.
3) ABSTRACT: Page 1, line 21
What was the hearing status of the study participants? Please specify in the methods section
4) INTRODUCTION: Page 1, line 39
I suggest adding that children can also suffer from tinnitus (I suggest citing the following article: Prevalence of tinnitus and hyperacusis in children and adolescents: a systematic review. BMJ Open. 2016 Jun 3;6(6):e010596. doi: 10.1136/bmjopen-2015-010596).
5) INTRODUCTION: Page 3, line 46
I suggest adding the important consideration that patients can suffer from multiple types of tinnitus at the same times, with different characteristics (I suggesting citing the following article: Tinnitus, Aural Fullness, and Hearing Loss in a Patient with Acoustic Neuroma and Pituitary Macroadenoma. Journal of Otorhinolaryngology, Hearing and Balance Medicine. 2023; 4(1):2. https://doi.org/10.3390/ohbm4010002)
6) MATERIALS AND METHODS: Page 3, line 132
Was hearing loss bilateral? Please clarify.
7) MATERIALS AND METHODS: Page 3, line 132
Which classification of hearing loss was used? Please clarify.
8) DISCUSSION: Page 17, line 808
I suggest expanding the limitation section (e.g. single center, qualitative analysis,..)
9) DISCUSSION: Page 17, line 808
Which are the future prospects of this study? Please provide a detailed description.
Author Response
Reviewer 2:
I would like to thank the authors for their submission and allowing me to review their work.
This is an interesting study on an important topic. However, I would be grateful if you could add further explanations and changes on the following points:
- Abstract: Page 1, line 19
Reply: Done
Please specify the mean age (standard deviation) and gender of the study population.
- Abstract: Page 1, line 21
Reply: Done
Where and when was the study conducted? Please specify in the methods section.
- Abstract: Page 1, line 21
Reply: Done
What was the hearing status of the study participants? Please specify in the methods section
Reply: The information was added line 200 : « A total of three students had hearing loss. One of them had mild unilateral hearing loss in the right ear, and two students had bilateral hearing loss (1 moderate, and 1 moderately severe). »
- Introduction: Page 1, line 39
I suggest adding that children can also suffer from tinnitus (I suggest citing the following article: Prevalence of tinnitus and hyperacusis in children and adolescents: a systematic review. BMJ Open. 2016 Jun 3;6(6):e010596. doi: 10.1136/bnjopen-2015-010596).
Reply: The suggestion was added in the modified manuscript of the Introduction as follows (line 42): « The prevalence of tinnitus is estimated around the world to be close to 15-20% of the general population [3–5]. The prevalence of tinnitus usually increases with age, with a higher prevalence for the older age groups. It is also known that tinnitus may occur at an earlier stage in life, such as in children and adolescents [6]. »
- Introduction: Page 3, line 46
I suggest adding the important consideration that patients can suffer from multiple types of tinnitus at the same times, with different characteristics (I suggesting citing the following article: Aldè, M., Pignataro, L., & Zanetti, D. (2023). Tinnitus, Aural Fullness, and Hearing Loss in a Patient with Acoustic Neuroma and Pituitary Macroadenoma. Journal of Otorhinolaryngology, Hearing and Balance Medicine, 4(1), 2.
Reply: The suggestion was added, as well as other citations as follows (line 55): It has been reported that many individuals may hear more than one tinnitus sound [2,9,10], and that multiple types of tinnitus can interact and reside within a single patient [11].»
[2] Stouffer, J.L., Tyler, R.S., 1990. Characterization of tinnitus by tinnitus patients. J. Speech Hear. Disord. 55, 439–453.
[9] Basile, C.-É.; Fournier, P.; Hutchins, S.; Hébert, S. Psychoacoustic Assessment to Improve Tinnitus Diagnosis. PLoS ONE 2013, 8, e82995, doi:10.1371/journal.pone.0082995.
[10] Norena, A., Micheyl, C., Che ry-Croze, S., Collet, L., 2002. Psychoacoustic characterization of the tinnitus spectrum: implications for the underlying mechanisms of tinnitus. Audiol. Neurootol. 7, 358–369.
[11] Aldè, M., Pignataro, L., & Zanetti, D. (2023). Tinnitus, Aural Fullness, and Hearing Loss in a Patient with Acoustic Neuroma and Pituitary Macroadenoma. Journal of Otorhinolaryngology, Hearing and Balance Medicine, 4(1), 2.
- Materials and methods: Page 3, line 132
Was hearing loss bilateral? Please clarify.
Reply: The mention of HL laterality was added in Table 1 and the manuscript was modified as follows: A total of three students had hearing loss. One of them had mild unilateral hearing loss in the right ear, and two students had bilateral hearing loss (1 moderate, and 1 moderately severe). The two students with moderate hearing loss wore hearing aids.
- Materials and methods: Page 3, line 132
Which classification of hearing loss was used? Please clarify.
Reply: The manuscript was modified under the section Material and Methods in the sub-section 2.2.1 Audiometry as follows: Hearing thresholds above 20 dB HL at any of the standard frequencies was considered a hearing loss. Hearing loss was then classified as follows: hearing losses between 20- and 40-dB HL were considered mild, 41- and 55-dB HL moderate, 56- and 70-dB HL moderately severe, 71 and 90 dB HL severe, and hearing loss above 91 dB HL was considered profound.
Discussion: Page 17, line 808
I suggest expanding the limitation section (e.g. single center, qualitative analysis,..)
Reply: Done, starting line 1037
- Discussion: page 17, line 808
Which are the future prospects of this study? Please provide a detailed description.
Reply: To address the suggestion made above, a new limitations/perspective section was added to the modified manuscript. See line 1037.
Round 2
Reviewer 1 Report
The replies of the Authors are complete and appropriate.